# Short-Chain Fatty Acids and Their Association with Signalling Pathways in Inflammation, Glucose and Lipid Metabolism

**DOI:** 10.3390/ijms21176356

**Published:** 2020-09-02

**Authors:** Jin He, Peiwen Zhang, Linyuan Shen, Lili Niu, Ya Tan, Lei Chen, Ye Zhao, Lin Bai, Xiaoxia Hao, Xuewei Li, Shunhua Zhang, Li Zhu

**Affiliations:** 1College of Animal Science and Technology, Sichuan Agricultural University, Chengdu 611130, China; hejin19960812@163.com (J.H.); sicau_zhangpeiwen@163.com (P.Z.); shenlinyuan0815@163.com (L.S.); dky9829@126.com (L.N.); tanya@stu.sicau.edu.cn (Y.T.); chenlei815918@sicau.edu.cn (L.C.); zhye3@foxmail.com (Y.Z.); blin16@126.com (L.B.); xiaoxia6363@126.com (X.H.); xuewei.li@sicau.edu.cn (X.L.); 2Farm Animal Genetic Resource Exploration and Innovation Key Laboratory of Sichuan Province, Sichuan Agricultural University, Chengdu 611130, China; 3Institute of Animal Husbandry and Veterinary, Guizhou Academy of Agricultural Science, Guiyang 550005, China

**Keywords:** short-chain fatty acids, inflammation, glycose and lipid metabolism, signalling pathways

## Abstract

Short-chain fatty acids (SCFAs), particularly acetate, propionate and butyrate, are mainly produced by anaerobic fermentation of gut microbes. SCFAs play an important role in regulating energy metabolism and energy supply, as well as maintaining the homeostasis of the intestinal environment. In recent years, many studies have shown that SCFAs demonstrate physiologically beneficial effects, and the signalling pathways related to SCFA production, absorption, metabolism, and intestinal effects have been discovered. Two major signalling pathways concerning SCFAs, G-protein-coupled receptors (GPRCs) and histone deacetylases (HDACs), are well recognized. In this review, we summarize the recent advances concerning the biological properties of SCFAs and the signalling pathways in inflammation and glucose and lipid metabolism.

## 1. Introduction

Short-chain fatty acids (SCFAs), also called volatile fatty acids, are organic linear carboxylic acids with fewer than six carbons, including acetic acid, propionic acid, butyric acid, and valeric acid. Among them, acetate (C2), propionate (C3), and butyrate (C4) are the most abundant (≥95%) [1], and the C2, C3, and C4 are in an approximate molar ratio of 60:20:20, respectively [2,3]. Except for a small portion of SCFAs obtained directly from food, most are produced by intestinal microbial anaerobic fermentation. Furthermore, around 500–600 mmol of SCFAs are produced in the intestinal tract per day, depending on the diet, type and number of microbiomes, and residence time in the intestinal tract [4,5].

Dietary fibre is the main food component that affects the production of SCFAs, which are mainly derived from plant foods. Humans lack enzymes that breakdown dietary fibre. Therefore, dietary fibre passes through the upper digestive tract largely undigested and is fermented in the caecum and large intestine by anaerobic microorganisms. A key mechanism of metabolic regulation by the gut microbiota is the production of SCFAs. Different intestinal microbes will produce different amounts of SCFAs. Bacteroidetes (Gram negative) mainly produce acetate and propionate, whereas Firmicutes (Gram positive) use butyrate as the primary metabolic end product [5]. Human-derived *Bifidobacterium breve* UCC2003 and *Bifidobacterium longum* NCIMB 8809 use novel oligosaccharides to produce acetate [6], and *Bifidobacterium animalis* subsp. Lactis GCL2505 can also increase the production of acetate [7]. Although anaerobic fermentation of fibre by intestinal microorganisms is the largest source of SCFAs, SCFAs are also formed as products from peptide and amino acid fermentation (less than 1%) [8,9]. Although diet and the microbiome are the main factors affecting the production of SCFAs, species evolution and colonic environment have important effects [10,11].

Colonocytes absorb SCFAs after they are produced mainly via H^+^-dependent or sodium-dependent monocarboxylate transporters [12]. After supplying colonocytes, the remaining SCFAs are transported through the blood to various parts of the body. These SCFAs can be used as substrates to synthesize sugars or lipids and can also be used as cytokines to regulate metabolism [13,14,15,16]. These results show that SCFAs are carried from the intestinal cavity into the blood vessels of the host and finally to organs as substrates or signalling molecules.

Additionally, a growing number of functions are attributed to SCFAs. For a long time, many studies have argued that dietary fibre and resistant starch have many benefits, such as reducing cholesterol levels and maintaining normal blood glucose levels [17,18,19], and these benefits of high-fibre diets are at least a part put down to SCFAs. When SCFAs are produced in the gut, many of them are used as energy sources. For humans, SCFAs provide approximately 10% of the daily calorie requirement [20]. SCFAs also improve gut health through several partial effects, including maintaining the integrity of the intestinal barrier, producing mucus, preventing inflammation and reducing the risk of colorectal cancer [21,22,23,24,25]. Furthermore, the effects on activating brown adipose tissue [26], regulating liver mitochondrial function [27], maintaining body energy homeostasis [28], controlling appetite [26] and sleep [29] are all related to SCFAs.

Among the many physiological functions of SCFAs, the regulation of inflammation and glucose and lipid metabolism, has concerned most researchers. SCFAs play a very important role in these physiological processes, but how do SCFAs work? This mini review aims to summarize the current knowledge concerning the biological properties of SCFAs and their signalling pathways in inflammation and glycose and lipid metabolism.

## 2. G Protein-Coupled Receptors (GPCRs)

G protein-coupled receptors (GPCRs), involving seven transmembrane domains, are the biggest receptor family in mammals and participate in regulating almost all cell and physiological functions in the body. GPCRs can bind chemicals in the extracellular environment, such as odourants, hormones, neurotransmitters, chemokines, sugars, lipids, and proteins. After being activated by a ligand, GPCRs can bind to four different heterotrimeric G proteins (G_s_, G_i_/_o_, G_q_/_11_ and G_12_/_13_), which can influence the activity of single or multiple effectors, such as second-messenger-producing enzymes or ion channels. Presently, GPR41 and GPR43 have been identified as the most important receptors of SCFAs in the GPCR family [30,31]. After the discovery of SCFA receptors, GPR41 was renamed free fatty acid receptor 3 (FFAR3) and GPR43 was renamed FFAR2. Different SCFAs and receptors have different affinities. Specifically, in humans, the affinity ranking of FFAR2 is C2=C3>C4>C5=C1 and that of FFAR3 is C3=C4=C5>C2>C1 [32,33]. Both FFAR2 and FFAR3 are associated with metabolic diseases, and they have become effective targets for the treatment of type 2 diabetes, asthma, cardiovascular disease, as well as metabolic syndrome.

FFAR2 is widely expressed in vivo. FFAR2 is present in pancreatic islet α and β cells [34,35] and intestinal enteroendocrine cells (I and L cells) (Table 1) [36,37]. FFAR2 expression has been identified along the entire gastrointestinal tract and white adipocytes [38,39]. Moreover, FFAR2 is present in monocytes, neutrophils, eosinophils, intestinal Treg cells and other immune cells [40,41,42]. An increasing number of studies have confirmed the function of FFAR2. Attilio et al. found that FFAR2 stimulates insulin secretion and reduces apoptosis in mouse and human islets in vitro [43]. Kendle et al. found that FFAR2 gene knockout mice showed significant aggravation in colitis, arthritis and asthma [40]. FFAR2 significantly inhibits the lipolysis of primary human fat cells and stimulates the secretion of Glucagon-Like Peptide-1 (GLP-1) by mouse enterocrine tumour (STC-1) cells [44]. However, in another study, FFAR2 inhibited intestinal transport, intestinal function, and food intake through the peptide YY (PYY, an enteroendocrine hormone that reduces gut motility) pathway, along with limiting the function of GLP-1 [45]. Thus, although SCFA may alter the function of some proteins, further analysis is needed to determine the specific role of SCFA in different situations.

Additionally, SCFAs can activate Gi/o protein through FFAR2 and inhibit adenylate cyclase, reducing the production of cAMP from ATP [46]. Activation of FFAR2 can phosphorylate ERK1/2, activate mitogen-activated protein kinase (MAPK) and increase Ca^2+^ concentration [47]. C3 can promote the release of anti-inflammatory interleukin 10 (IL-10) from Treg cells, a process that occurs in a manner specific to FFAR2 [42]. Moreover, SCFAs can inhibit the expression of IL-6, IL-1β and tumour necrosis factor α (TNFα) to exert anti-inflammatory effects by FFAR2 [48,49,50].

In addition to intestinal enteroendocrine cells, FFAR3 was found in K cells, enteric neurons and sympathetic ganglia (Table 1) [51,52]. Consistent with FFAR2 function, FFAR3 also plays important roles in inflammation. C3 can inhibit the expression of IL-4, IL-5, and IL-17A through FFAR3, and C4 can inhibit the expression of induced nitric oxide synthase (iNOS), TNFα, IL-6, and monocyte chemoattractant protein-1 (MCP-1) [53,54,55]. Both FFAR2 and FFAR3 can be combined with G_i/o_ receptor [31,56,57]. But FFAR3 can only function through G_i/o_ receptor, while FFAR2 is pleiotropic, and it can also perform its functions through the G_q/11_ pathway [30].

Certainly, FFAR3 also has different functions from FFAR2. FFAR3 is mainly expressed in neurons with a vasoconstrictor phenotype [58]. Many studies have investigated the role of FFAR3 in vasoconstriction. For example, C3 causes protective effects on allergic airway inflammation through FFAR3 [59]. FFAR3 activation induces vasodilation and lowers systemic blood pressure in vascular smooth muscle cells [60,61]. In human airway smooth muscle (ASM), FFAR3 promotes ASM contraction by reducing cAMP and increasing intracellular Ca^2+^ [62]. FFAR3 does not only play a role in vasoconstriction. Ørgaard et al. found that SCFA treatment did not affect the secretion of glucagon in vitro, and the inhibitory effects on insulin secretion were weak but induced a strong increase in somatostatin secretion [63]. In another study, FFAR3 signalling mediated glucose-stimulated insulin secretion through the Gi/o-sensitive pathway, and FFAR3 signalling negatively mediates insulin secretion [64]. Moreover, exogenous supplementation of SCFAs can reduce liver fat content and improve liver metabolism by inhibiting the expression of lipid synthesis genes in FFAR3-deficient mice liver but not FFAR2-deficient [65].

In initial reports, GPR109A was found in the intestinal tract, partial immune cells, and adipocytes (Table 1) [66,67]. Interestingly, these researchers found that GPR109A is not activated by SCFAs but niacin, a famous lipid-lowering substances [68]. Almost simultaneously, the receptor GPR109b, which is homologous to the GPR109A sequence, was also found, but it had a lower affinity for niacin [69]. Over time, researchers found that SCFAs also activate GPR109A. Unlike FFAR2 and FFAR3, GPR109A is activated by longer SCFAs, mainly C4 [70]. GPR109A is also widely expressed in vivo. GPR109A was detected in the bone marrow, lymph node, prostate and spleen by qRT-PCR and plays an important role in these organs [71]. Similarly, GPR109A is expressed in both mouse and human islet β cells, although very rarely. And GPR109A, also expressed in breast tissue, inhibits the progression of breast tumours by promoting apoptosis [72]. In addition, the intestinal epithelial cells are also the site of GPR109A expression [67]. Moreover, GPR109A is also expressed in the brain, especially at the ventrolateral end of the rostral medulla, and is related to blood pressure regulation [73]. Li et al. found that GPR109A functions as an anti-inflammatory effector by inhibiting the Akt/mTOR signalling pathway in MIN6 pancreatic β cells [74].

The GPR109A receptor has many functions. For example, Kaye et al. found that SCFAs can exert cardiovascular protection through GPR43/GPR109A receptors and can affect DNA methylation to increase the number of Treg cells [75]. Additionally, GPR109A inhibits insulin secretion and is downregulated in pancreatic beta cells of type 2 diabetes [76]. Activation of niacin receptor HCA2 can suppress macrophage migration induced by chemical kinases [77]. These effects were mainly caused by the effect of activated GPR109A on the intracellular CAMP content. In islets, GPR109A activation reduces insulin secretion. Interestingly, GPR109A expression was significantly reduced in the islets of db/db mice in the same study [76]. In adipose tissue, lipase activity, plasma triglyceride levels and free fatty acid levels were reduced following GPR109A activation. Moreover, in the gut, GPR109A was used as a nutrient signal by enhancing the expression of MCT-1 through C4 treatment. GPR109A KO mice were more susceptible to dss-induced colitis than the control group, and GPR109A was found to be downregulated in human colon cancer cell lines (along with mouse models of colon cancer) [42,67,78]. In summary, GPR109A is expressed in many organs and cells and is beneficial to metabolism.

## 3. Histone Deacetylases (HDACs)

Histone deacetylase (HDAC) is a type of protease that plays an important role in chromosome structure modification and gene expression regulation. In general, histone acetylation is conducive to the dissociation of DNA from histone octamers and the relaxation of nucleosome structure, causing various transcription factors and synergistic transcription factors to specifically bind to DNA binding sites and activate gene transcription. Within the nucleus, histone acetylation and histone deacetylation are in dynamic equilibrium and are jointly regulated by histone acetyltransferase (HAT) and histone deacetylase (HDAC). HAT transfers the acetyl group of acetyl-CoA to a specific lysine residue at the amino terminus of histones. HDAC deacetylates histones, making them bind to negatively charged DNA, curling chromatin, and inhibiting gene transcription.

SCFAs are natural inhibitors of HDACs. The inhibitory effect of SCFAs on HDACs depends on the SCFA concentration, and the inhibitory effect of a high SCFA concentration is more obvious. Among all SCFAs, butyrate is the most potent inhibitor of HDAC activity. Studies dating back to 1978 have shown that SCFAs can increase histone acetylation. Sealy and Chalkley found that treatment of hepatoma tissue with acetate, propionate, or butyrate leads to a global increase in histone acetylation [79]. In the same year, Boffa et al. found that butyrate had stronger HDAC inhibitory activity than propionate in HeLa cells and colon cancer cell lines [80]. Similarly, in the follow-up study, Waldecker et al. found that HDAC was inhibited at concentrations ≥ 10 mM in most SCFAs tests, except for acetate, while butyrate showed significant inhibition at 2 mM and almost 40% inhibition at 1 mM; butyrate also showed better inhibition in nuclear extracts [81]. However, in both studies and some other studies, acetate had little or no effect on HDAC. However, this lack of effect on HDACs by acetate may be tissue dependent since others have shown that acetate can inhibit HDACs. In 2011, Soliman and Rosenberger found that exogenous supplementation significantly reduced HDAC levels in rat brain and liver without affecting HAT levels [82]. On the other hand, Bulusu et al. found that acetate can be converted to acetyl-coa directly providing the acetyl group for histone acetylation but in a very small proportion [83]. The role of acetate in histone acetylation is complex because it can supply acetyl units for HAT and act as an HDAC inhibitor.

Although the mechanism by which SCFA inhibits HDACs is still unclear, SCFAs may directly act on HDACs through transporters into the cell, or may indirectly act on HDACs through GPCR activation. For example, C4 inhibits the production of nitric oxide and inflammatory cytokines such as IL-6 and IL-12 induced by lipopolysaccharide, and the way does not depend on GPCRs, presumably by inhibiting HDAC [84]. In this study, the author believes that C4 acts directly on HDAC. SCFAs can enter cells through the transporter sodium coupled to monocarboxylic acid transporter 1 (SMCT-1) without passing through the membrane receptor, occupying the active site of HDACs and causing inhibition [85]. In another report, Wu et al. found that activation of GPR41 can inhibit histone acetylation in Chinese hamster ovary cell lines by inhibiting HDAC [86]. Additionally, GPR41, GPR43 and GPR109 may be involved in SCFA-mediated HDAC inhibition. How SCFAs directly or indirectly inhibit HDAC activity remains unclear, and extensive research is needed to answer these questions.

Generally, HDAC inhibition promotes chromatin acetylation and target gene transcription, thereby affecting cell function, although it is not clear how cell and gene selectivity is achieved. Additionally, HDAC inhibition has numerous downstream consequences. Our understanding of how SCFAs inhibits HDAC is still in its preliminary stages and will require us to study and explain this question in the future.

## 4. SCFAs and Inflammation

LPS (lipopolysaccharide), which comprises lipids and polysaccharides, is a major component of the cell wall of Gram-negative bacteria. As a classical pattern-recognition molecule, LPS plays an important role in the dynamic process of natural immunity. The LPS levels are one of the diagnostic markers in inflammatory diseases [87]. After LPS falls off the bacterial cell wall, it is detected by lipopolysaccharide binding protein (LBP) in the serum and applied to the TLR on macrophages or neutral cells. These cells then secrete pro-inflammatory factors such as TNF, IL-1β and IL-6, leading to an inflammatory response [88,89]. The two major inflammation-related signalling pathways in cells, NF-κB and MAPK pathways, are activated after TLR activation. NF-κB belongs to a family of nuclear transcription factors comprising p50, p52, REL, REL-a, and REL-B, and it is the primary response to harmful cellular stimuli. Currently, several downstream mediators have been identified to activate the NF-κB pathway: TNF-α, IL-1, IL-2, IL-6, IL-8, IL-12, iNOS, COX2, chemokines, adhesion molecules, and colony stimulating factors [90,91,92,93]. The MAPK signalling pathway is involved in the regulation of various key functions in the body and plays an important role in cell proliferation, differentiation and apoptosis. Studies have shown that inhibition of the MAPK pathway in the lung can abolish the LPS-induced TNF production, inhibit the recruitment of neutrophils in pulmonary bronchi, and completely disable the endotoxin-induced inflammatory response [94]. Presently, three downstream MAPK pathways have been identified: ERK1/2, JNK/ASPK and P38 MAPK. Activation of the LPS-induced ERK pathway results in the secretion of inflammatory cytokines, such as TNF-α and IL-6, and increased expression of iNOS and nitric oxide (NO) [95]. Similarly, the JNK and P38 MAPK pathways also play important roles in inflammation. When the body has an inflammatory response, JNK can induce apoptosis and promote the activation of inflammatory factors TNF-α and IL-1β [96]. Inhibition of the p38/MAPK signalling pathway in airway smooth muscle plays an anti-inflammatory role by reducing the secretion of the inflammatory factor TNF-α.

Interestingly, although LPS can induce inflammation through different signalling pathways, many studies have shown that SCFAs can inhibit LPS-induced inflammation through the above GPCRs and HDAC (Figure 1). For example, butyrate and propionate reduce the expression of TNF and nitric oxide synthase (NOS) in LPS-induced monocytes [97]. Furthermore, butyrate treatment can induce the phosphorylation of ERK, p38, JNK and NF-κB p65 through TLR4 in colon cancer cells [98]. Acetate restrains LPS-induced TNFα secretion from mice and human mononuclear cells by activating FFAR receptors [99]. In other studies, butyrate and propionate treatment can inhibit the secretion of TNFα and the activity of NF-κB and up-regulate the expression of anti-inflammatory factors IL-10 in LPS-activated mononuclear cells and neutrophils by HDAC inhibition [84,100,101]. Additionally, SCFAs can down-regulate the expression of IL-8 in airway inflammation by activating FFAR2 and FFAR3 receptors [102]. In macrophages, butyrate exerts anti-inflammatory effects by reducing the production of iNOS, TNFα, MCP-1 and IL-6 through FFAR3 receptor [53], and cytokines such as IFN-γ can increase GPR109A expression [103]. These anti-inflammatory effects are attributed to the activation of FFAR2, FFAR3 and GPR109A receptors; however, whether LPS induction affects the inflammatory responses has not been directly described.

Although it has been widely reported that SCFAs play anti-inflammatory roles through different signalling pathways, some studies have found that SCFAs also play pro-inflammatory roles through GPCRs. For example, Kim et al. found that SCFAs activate GPR41 and GPR43 on intestinal epithelial cells to protect immunity and tissue inflammation in C57BL6 mice without FFAR2 or FFAR3 receptors [104]. The cause may be due to the authors using different SCFAs to activate the receptor. In this article, ethanol as SCFA may not be desirable. Strictly speaking, acetic acid or acetate is the appropriate experimental material to study SCFA.

## 5. SCFAs and Glucose Metabolism

Glucose is phosphorylated by hexokinase to form glucose 6-phosphate, which enters the glycolysis pathway and produces ATP for energy supply or glycogen for energy storage. The liver and skeletal muscles are important tissues for glucose metabolism. The regulation of blood glucose balance includes two aspects: the generation of blood glucose (feeding and gluconeogenesis) and consumption of blood glucose (exercise consumption and glycogen storage). The blood glucose concentration is regulated by complex hormones, among which the main ones are glucagon secreted by islet α cells and insulin secreted by islet β cells. Studies have shown that the activation of FFAR3 by SCFA can stimulate the secretion of intestinal hormone PYY in endocrine cells, enhancing the absorption of glucose in muscle and adipose tissue and producing a sense of satiety and reduced food intake (Figure 2) [105,106,107]. Moreover, SCFA can stimulate the secretion of glucagon-like peptide-1 (GLP-1) by activating FFAR2, which indirectly regulates blood glucose levels by increasing insulin secretion and decreasing pancreatic glucagon secretion [108,109]. In addition to glucagon and insulin, leptin also plays an important role in blood glucose balance. Leptin is an aliphatic factor secreted by adipocytes and mainly regulates food intake, body weight and energy metabolism through the central nervous system. Sakakibara et al. found that SCFAs can increase leptin secretion by activating FFAR2 in vivo or in vitro [110]. Fujikawa et al. found that leptin promotes glucose uptake in brown adipose tissue and the soleus muscle and improves liver metabolism by acting on gamma-aminobutyric acid (GABA) and optic nerve melanocortin (POMC) neurons in the hypothalamus [111]. Additionally, leptin can directly promote the synthesis of liver glycogen and muscle blood glucose uptake [112,113].

Furthermore, SCFAs have functions other than regulating blood sugar balance through hormones. Many glucose transporters are found on the cell. GLUT4, a speed-limiting protein that allows glucose to enter the cell, is mainly found in skeletal muscle cells. Studies have shown that short-chain fatty acids can increase the expression of GLUT4 and translocate it to the cell membrane, promoting the absorption of more glucose by myoblasts [114]. In the same study, the authors also found that SCFAs phosphorylated AMPK and its downstream product ACC. Hong et al. found that butyric acid increased the phosphorylation of AMPK in high-fat-fed mice [115]. SCFAs directly activate AMPK by increasing the ratio of AMP/ATP [116]. In skeletal muscle, AMPK activation inhibits glycogen and protein synthesis and promotes glucose transport and fatty acid oxidation [117,118]. In the liver, AMPK activation reduces the gene expression of glucose 6-phosphatase (G6Pase) and phosphoenolpyruvate carboxy-kinase (PEPCK), which are the key enzymes of gluconeogenesis [110]. Similar results were found by Kondo and Gao et al. In their study, SCFA directly activated the AMPK signalling pathway by increasing the AMP/ATP ratio in the muscles and liver [13,119]. AMPK can affect many downstream signalling pathways, such as mTOR, PGC-1A and FOX0_3_ [120,121,122]; however, whether these signalling pathways are related to glucose metabolism activated by SCFA remains to be further studied.

## 6. SCFAs and Lipid Metabolism

In three generations of substance metabolism, lipid metabolism is a key process that short-chain fatty acids can regulate. As members of the fatty acid family, SCFAs provide a substrate for lipid synthesis. Zambell et al. proved that acetate and butyrate are the main synthetic lipid substrates in rat colonic epithelial cells, which convert SCFAs into acetyl-CoA [123]. Similarly, Kindt et al. found that the intestinal microbiota promotes liver fatty acid metabolism by providing a high level of acetate as a precursor to the synthesis of palmitate and stearate [124]. Acetyl-CoA can not only enter the tricarboxylic acid cycle to generate energy but also generate palmitic acid under the action of the cytosolic enzyme system, which can transfer to mitochondria to extend the carbon chain and form triglycerides with other substances stored in adipose tissue. SCFAs are not only involved in lipid metabolism as a substrate but also can be used as a regulatory factor to regulate lipid metabolism. The experiment of Li et al. showed that butyric acid increased the oxidation of fatty acids in brown adipose tissue and improved the obesity and insulin resistance caused by diet [26]. Butyric acid can also promote the browning of white tissue, reduce the size of adipose cells morphologically, and increase the number of multicellular adipose cells [13].

In addition to glucose metabolism, AMPK signalling also plays a role in lipid metabolism (Figure 3). Previous studies have shown that AMPK activation increases PGC-1α expression in adipose tissue and skeletal muscle [125,126,127,128]. Additionally, PGC-1α regulates the transcription activity of various transcription factors, including peroxisome proliferator-activated receptor α (PPARα) and peroxisome proliferator-activated receptor γ (PPARγ) [13,129,130,131]. Huang et al. found that activation of the AMPK signalling pathway in db/db mice increased the expression of PPARα and p-ACC in the liver, reducing the levels of triglycerides and free fatty acids [132]. Unfortunately, SCFAs were not used as an activator of AMPK in this study. However, in another study, the authors also found consistent results after activating AMPK using SCFAs [114]. Additionally, hormone-sensitive lipase (HSL) and adipose triglyceride lipase (ATGL), as the main enzymes of lipolysis, are regulated by AMPK. Studies have shown that the activated AMPK signalling pathway can promote the expression of HSL and ATGL and promote lipolysis [133,134,135]. However, other studies have reported different results. Houslay et al. found that SCFA inhibited adenylate cyclase, reducing cAMP production by ATP and protein kinase A (PKA) activity [46]. Subsequent studies have shown that reduced PKA activity leads to dephosphorylation and inactivation of HSL in adipose tissue [136]. Similarly, Jocken et al. found that acetate played an anti-lipolysis role by inhibiting HSL phosphorylation activity in human multipotent adipose tissue-derived stem cells [137]. Reducing the plasma levels of free fatty acids by inhibiting intracellular lipolysis may also regulate lipid metabolism by SCFA. Moreover, the activation of AMPK in rat liver cancer cells inhibited the synthesis of liver fatty acids by inhibiting sterol regulatory element-binding protein 1C (SREBP-1c, the main regulator of liver adipogenic gene expression) [138]. Furthermore, PGC-1α regulates cholesterol, lipid and sugar metabolism. In summary, AMPK activation triggers the expression of PGC-1α, and SREBP-1c is a potential target.

The uncoupling protein (UCP) plays an important role in lipid metabolism. There are three main subtypes in adipose tissue—UCP1, UCP2, and UCP3—which limit ATP synthesis, increase thermogenesis, and allow fatty acid oxidation to reduce lipid deposition. Gao et al. found that SCFA can increase the protein expression of PGC-1 and UCP-1 in brown adipose tissue [13]. Besten et al. found that SCFAs stimulate mitochondrial fatty acid oxidation by activating the UCP2-AMPK-ACC signalling pathway in human HepG2 hepatocytes and mouse 3T3L1 adipocytes [139]. Hong et al. found that butyrate upregulated the expression of UCP2, UCP3 and fatty acid oxidase in skeletal muscle. After butyrate administration, histone markers H3K9Ac with high expression of gene activation were detected in the promoter regions of adiponectin receptor 1/2, Ucp2 and Ucp3 in the muscle of obese mice [115]. Thus, butyrate not only increases heat production and lipid consumption through UCP but also improves lipid metabolism by activating adiponectin. Other genes, such as leptin, may also play a role in the regulation of lipid metabolism by SCFA, a topic that is not discussed too much here.

## 7. Summary

Overall, SCFAs have been shown to benefit many aspects of the body’s metabolism. SCFAs have become a hot topic among researchers in terms of their initial food intake, intestinal flora composition and subsequent regulation of metabolism. Although many studies have investigated the role of SCFAs in inflammation, glucose metabolism and lipid metabolism, no systematic review exists to elucidate the SCFA-related signalling pathway in inflammation, glucose metabolism and lipid metabolism. Thus, we hope that this review increases awareness of the dominant roles of SCFAs in inflammation, glucose and lipid metabolism.

In studies of inflammation, although some studies have shown that SCFA is detrimental, most studies have shown that SCFA can reduce inflammation. However, in the existing studies, we found that butyrate was favoured by most researchers in the studies on SCFA-associated inflammation; few reports concerned other SCFAs. Thus, the inhibitory effect of different SCFAs in the same inflammatory model must be studied. The role of SCFAs in inflammation requires further study. Additionally, the regulation of glucose metabolism by SCFAs is mainly achieved by maintaining blood glucose stability. The effects of insulin and glucagon secretion are crucial for regulating glucose metabolism. Regarding lipid metabolism, SCFAs can increase fatty acid oxidation, inhibit fatty acid synthesis, increase heat production and reduce fat storage.

SCFAs have demonstrated a strong ability to regulate metabolism, but the regulatory network of SCFAs is very complex and the underlying molecular mechanism remains unclear. And the current study still has limitations. For example, although mice and other laboratory animals are used as model animals to study whether it is applicable to humans, but whether these animal data are fully applicable to humans is still a question worth exploring. However, studies in humans are mainly carried out using available mature cell lines or analysing SCFAs residue in faeces, and the inability to obtain data in vivo is still a big problem. Most of the current studies have focused on butyrate; whether it fully represents SCFAs is questionable. Although exogenous supplements can improve metabolism, could they be used in humans? Currently, there are no mature SCFA products on the market. Follow-up studies should focus on the development of effective SCFAs for clinical use.

## Figures and Tables

**Figure 1 ijms-21-06356-f001:**
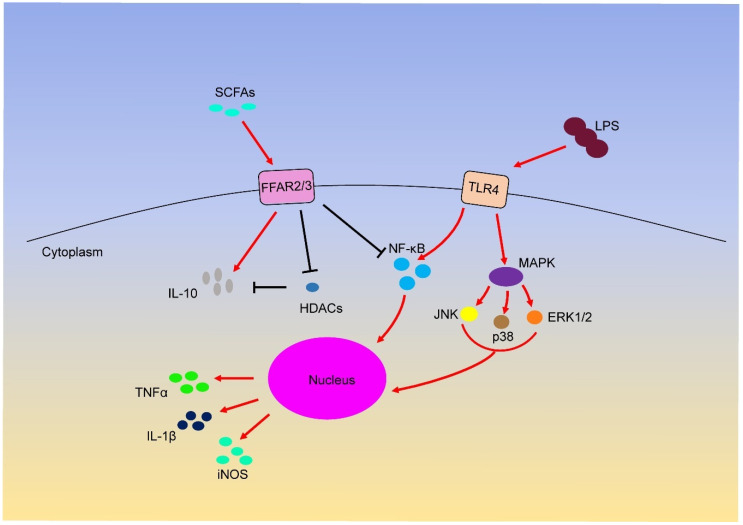
Short-chain fatty acids (SCFAs) regulates inflammation through FFAR2/3 receptor.

**Figure 2 ijms-21-06356-f002:**
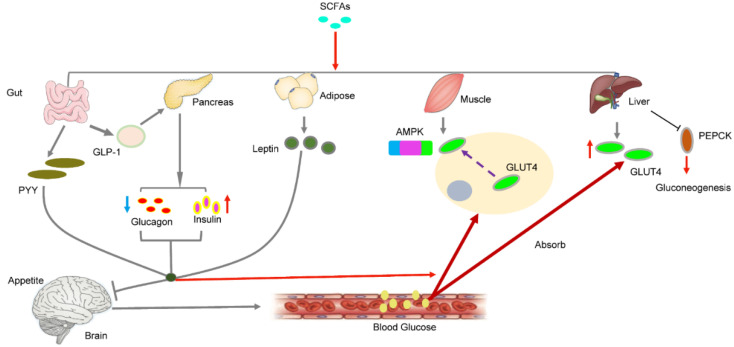
SCFAs regulates blood sugar by regulating the secretion of hormones in different tissues.

**Figure 3 ijms-21-06356-f003:**
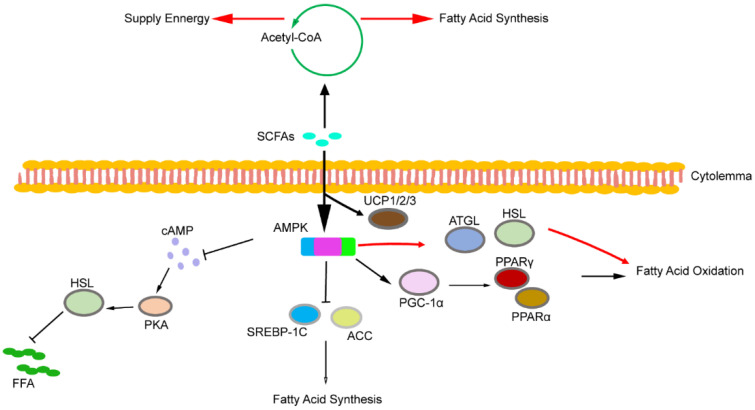
SCFAs regulates lipid metabolism by increasing fatty acid oxidation, reducing lipid deposition and heat production.

**Table 1 ijms-21-06356-t001:** Short-chain fatty acids (SCFAs) receptor and its main expression site.

SCFAs Receptors	Alternative Names	G-Protein Coupling	Affinity	Expression	References
FFAR2	FFA2; GPR43	Gq/G_11_; Gi	C2=C3>C4>C5=C1	white adipocytes, immune cells, islet α and β cells, intestinal enteroendocrine cells	[34,35,36,37,38,39,40,41,42]
FFAR3	FFA3; GPR41	Gi	C3=C4=C5>C2>C1	immune cells, enteric neurons and sympathetic ganglia, intestinal enteroendocrine cells, islet α and β cells,	[34,35,36,37,51,52]
GPR109A	HCA2	Gi	C4, niacin	Macrophages, immune cells, Adipocytes, β cells microvascular endothelial cells, microglial cells	[66,67,68,69,70,71,72,73]

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
