# Peer review of "Short-Chain Fatty Acids and Their Association with Signalling Pathways in Inflammation, Glucose and Lipid Metabolism"

_ijms, 2020, doi:10.3390/ijms21176356_

Round 1

Reviewer 1 Report

The manuscript entitled „A review of short-chain fatty acids and the association with the signal pathways in inflammation and glycolipid metabolism “ by Jin He and coworkers reviews the current findings in the interplay between SCFA and inflammation, glucose and lipid metabolism under special consideration of the signaling pathways.

The authors start with an introduction about origin and overall function of SCFA. Signaling via G-protein-coupled receptors and histone deacetylases were described. In the main part, the authors summarize recent findings of the association between SCFA and inflammation, glucose and lipid metabolism. In the end, they try to give an outlook.

The review covers a wide range of influences SCFA might have. The paper could be a valuable contribution in understanding the role of SCFA in health and disease. After some modifications it is worth to be published in the International Journal of Molecular Sciences.

Please find the concerns in detail:

  • Introduction: page 2 line 12 “the genetic background of the host and the colonic environment have important effects” there is no citation for this hypothesis;
  • The manuscript is a little bit too bulky. For example, please skip the first page of chapter 4. It is not the aim of this review to explain what an inflammation is. Just introduce shortly what LPS means. For chapter “SCFA and glucose metabolism” it is the same. The first 12 lines can be omitted. Line 20: “ absorbed by organs” ???
  • The numbering is confusing. It ends with chapter 4. The following chapters are unnumbered.
  • The figures are a nice illustration of SCFA signaling. The sketch of SCFA implicates, however, a chain of 6 C-atoms. This is wrong. Is it possible to illustrate the SCFA structure in a more appropriate drawing?
  • In chapter “SCFA and glucose metabolism” some important recent publications are missing: (Kindt et al., 2018, Jocken et al., 2017). Kindt et al. found out that acetate is an important precursor for the synthesis of fatty acids and phospholipids in the liver.
  • Outlook: the English is poor and it is hard to read. You mention inconsistencies in the studies. Which? I could not find so many contradictions in your reviewed publications. You ask fo exogeneous supplementation. Yes, there are a lot of studies with supplementation of butyrate. However, this would be another literature review.
  • I would prefer a Summary instead of an Outlook. A good review attempts to summarize or synthesize what has been written on a particular topic but does not seek generalization or cumulative knowledge from what is reviewed. Instead, the review authors should undertake the task of accumulating and synthesizing the literature to demonstrate the value of a particular point of view. The review should inspire research ideas by identifying gaps or inconsistencies in a body of knowledge, thus helping researchers to determine research questions or formulate hypotheses.
  • The English is sometimes poor. For example page 4 line 17 (What do you mean regarding FFAR2 and FFAR3?) page 12 “soles muscle”?, page 13 “de node formation”-do you mean de-novo formation? Page 13 ”SCFAs can alter lipid metabolism as regulatory factors not the substrate”-what do you mean?
  • Please let a native speaker check the English language

Jocken, J. W. E., González Hernández, M. A., Hoebers, N. T. H., van der Beek, C. M., Essers, Y. P. G., Blaak, E. E. & Canfora, E. E. (2017). Front Endocrinol (Lausanne) 8, 372.

Kindt, A., Liebisch, G., Clavel, T., Haller, D., Hormannsperger, G., Yoon, H., Kolmeder, D., Sigruener, A., Krautbauer, S., Seeliger, C., Ganzha, A., Schweizer, S., Morisset, R., Strowig, T., Daniel, H., Helm, D., Kuster, B., Krumsiek, J. & Ecker, J. (2018). Nat Commun 9, 3760.

Author Response

Dear Reviewer:

Thank you very much for taking the time to review and comment our manuscript titled “A review of short-chain fatty acids and the association with the signal pathways in inflammation and glycolipid metabolism” (Manuscript ID: Ijms-880444). Your comments are important to the revision and quality improvement of the paper. After receiving the email, we carefully read the comments and suggestions, and tried our best to make detailed modifications in the corresponding positions of the paper. we marked red on major revisions. The above modification has been agreed by all authors of this paper. And we resubmit our manuscript and hope you could consider this paper to publish on International Journal of Molecular Sciences.

Thank you again for your careful review of this manuscript. If there is something that we are not doing properly, please give us guidance.

All the best with your work!

All authors

2020.8.14

According to your reviewer's suggestions and comments, the following contents are our modifications and replies:

Comment 1:

Introduction: page 2 line 12 “the genetic background of the host and the colonic environment have important effects” there is no citation for this hypothesis;

Response: We are very sorry for the lack of references in some places when writing the manuscript. We carefully checked these missing references and added them one by one. Here we have listed two documents for reference. In the tenth reference, we can learn that the pH of the colonic lumen varies with anatomical site and microbial fermentation of dietary residue. The eleventh reference analyzed the intestinal 16s rRNA microbiome data set of different animals (including Mammalia, Aves, Reptilia, Amphibia, and Actinopterygii) and found that the intestinal microbiome was significantly different. This leads to the difference in the production of SCFAs. These papers showed that the host’s genetic background and intestinal environment will affect the production of SCFAs.

Comment 2:

The manuscript is a little bit too bulky. For example, please skip the first page of chapter 4. It is not the aim of this review to explain what an inflammation is. Just introduce shortly what LPS means. For chapter “SCFA and glucose metabolism” it is the same. The first 12 lines can be omitted. Line 20: “absorbed by organs”???

Response: Thank you very much for your criticism and suggestions. It is true that some of the language descriptions in this manuscript are too bulky, we have simplified these parts hoping to achieve the effect of getting straight to the topic. Following the suggestion, the explanation of inflammation in section 4 was deleted. We directly lead to LPS and use this as an example to explain how SCFAs affects inflammation. And we also made the same modification in Section 5. We deleted the first 12 lines. About the line 20 of being absorbed by organs, I think it was caused by the improper use of our language. This sentence has now been revised to “These results indicate that SCFAs are transported from the intestinal lumen into the blood compartment of the host and finally to organs as substrates or signaling molecules.”

Comment 3:

The numbering is confusing. It ends with chapter 4. The following chapters are unnumbered.

Response: We are very sorry for the number confusion. We have added the subsequent numbers one by one. The current number has been changed to “1. Introduction, 2. G protein-coupled receptors (GPCRs), 3. Histone Deacetylases (HDACs), 4. SCFAs and Inflammation, 5. SCFAs and Glucose metabolism, 6. SCFAs and lipid metabolism and 7. Summary”.

Comment 4:

Outlook: the English is poor and it is hard to read. You mention inconsistencies in the studies. Which? I could not find so many contradictions in your reviewed publications. You ask foexogeneous supplementation. Yes, there are a lot of studies with supplementation of butyrate. However, this would be another literature review.

I would prefer a Summary instead of an Outlook. A good review attempts to summarize or synthesize what has been written on a particular topic but does not seek generalization or cumulative knowledge from what is reviewed. Instead, the review authors should undertake the task of accumulating and synthesizing the literature to demonstrate the value of a particular point of view. The review should inspire research ideas by identifying gaps or inconsistencies in a body of knowledge, thus helping researchers to determine research questions or formulate hypotheses.

Response: Thank you very much for your criticism. Your criticism is the source of our progress. To improve the English of the manuscript, we revised the languages and grammars of this manuscript by professional proofreading service. Indeed, there is almost no mention of contradictions in this article. The pro-inflammatory effects of SCFAs mentioned in citation 105 are not contradictory. The reason for this may be that the author used unconventional SCFAs. It’s true that I didn’t mention too much about external sources, as you said, it will be another review. We have made major changes in chapter 7, and we adjusted the outlook to a summary. We summarized the role of SCFAS in inflammation, glucose metabolism and lipid metabolism, and discussed the current problems, hoping to provide ideas for follow-up research. Hope our modification can meet your expectations. Thanks again for your suggestions!

Comment 5:

The English is sometimes poor. For example page 4 line 17 (What do you mean regarding FFAR2 and FFAR3?) page 12 “soles muscle”?, page 13 “de node formation”-do you mean de-novo formation? Page 13 “SCFAs can alter lipid metabolism as regulatory factors not the substrate”-what do you mean?

Please let a native speaker check the English language.

Response: Thank you very much for reminding. Following your requirement, we have revised the languages and grammars of manuscript by professional proofreading service. We hope the current language can clearly express what we want to say. The page 4 line 17 has been modified to “Certainly, FFAR3 also has different functions from FFAR2”. The page 12 “soles muscle” has been modified to “soleus muscle”. We are very sorry about our spelling mistake. The page 13 “de node formation” is consistent with what you mean, and this sentence was revised to “Zambell et al. proved that acetate and butyrate are the main synthetic lipid substrates in rat colonic epithelial cells, which convert SCFAs into acetyl-CoA”. AndSCFAs can alter lipid metabolism as regulatory factors not the substrate” has been modified to “SCFAs are not only involved in lipid metabolism as a substrate but also can be used as a regulatory factor to regulate lipid metabolism”. Thank you again for every suggestion.

Comment 6:

The figures are a nice illustration of SCFA signaling. The sketch of SCFA implicates, however, a chain of 6 C-atoms. This is wrong. Is it possible to illustrate the SCFA structure in a more appropriate drawing?

Response: Thank you very much for your suggestion. Indeed, six C-atoms do not represent SCFAs. In our view, this is just a symbol representing SCFAs, and does not mean that this is a structure of SCFAs. We all know that SCFAs are composed of one carboxyl group and different numbers of methyl groups. But we did not find a structure that can completely replace SCFAs. In order not to mislead other researchers, the picture has been modified.

Comment 1:

In chapter “SCFA and glucose metabolism” some important recent publications are missing: (Kindt et al., 2018, Jocken et al., 2017). Kindt et al. found out that acetate is an important precursor for the synthesis of fatty acids and phospholipids in the liver.

Response: Thank you very much for reminding. We carefully read the two references you mentioned. Kindt et al. found that the intestinal microbiota promotes liver fatty acid metabolism by providing a high level of acetate as a precursor to the synthesis of palmitate and stearate. And Jocken et al. found that acetate played an anti-lipolysis role by inhibiting HSL phosphorylation activity in human multipotent adipose tissue-derived stem cells. The two references have been added to the corresponding positions and marked in red. Thanks again for your reminder.

Once again, thanks for your careful reading and professional evaluation.

Reviewer 2 Report

The review by He and colleagues aims to pull together a very complex field explaining the role of short chain fatty acids in immune and metabolic regulation. The long-list of potential mechanisms is reported but perhaps less critically that what is needed to dissect what is relevant to animal studies and what is relevant to humans.

I am not sure that glycolipid in the title implies the same as those involved in glycolipid research would expect. Perhaps the title “A review of short chain fatty acids and the association with the signal pathways in inflammation, glucose and lipid metabolism” is more apt?

Section Comments

Introduction

Isobutyric and isovaleric acids are generally not include in the definition of SCFA. SCFA refer to straight chain fatty acids whereas isobutyric and isovaleric are usually defined as branched chain fatty acids.

It is not quite correct that dietary fibre passes through the upper digestive tract unaffected – there are numerous physico-chemical changes that occur – it would be more accurate to say that dietary fibre passes through the upper digestive tract largely undigested.

The contribution of amino acid fermentation (as opposed to metabolism which could also mean uptake by the microbiota) will depend on dietary intake and the introduction should better reflect that the contribution of protein derived amino acids to SCFA production depends on the protein : fibre ratio of the diet.

The statement “Although diet and microbiome are the main factors affecting the production of SCFAs, whereas the genetic background of the host and the colonic environment have important effects.” should be supported by references

SCFAs are absorbed by colonocytes after them were produced which mainly via H+-dependent or sodium-dependent monocarboxylate transporters. – needs references.

SCFAs are metabolized in the colonocytes and surplus part are transported into the portal circulation which used as an energy substrate for hepatocytes – not strictly true – most colonic acetate is oxidised in the peripheral muscle and main sequestration in the liver for acetate is into FA synthesis. Propionate is mostly oxidised in the liver.

In addition, besides the liver received some SCFA through the portal circulation, in which other organs and tissues are also affected.- this sentence does not make sense.

G-protein coupled receptors (GPCRs)

This section lists a long-list of potential signalling effects and mechanisms of SCFA through FFAR2, FFAR3 and GPR109A and interchanges between animal and human evidence quite seemlessly. However, can the animal data simply be extrapolated to humans given that there may be species differences in GPCR signalling? This section would really benefit from a critical analysis of

- what do we know from cell studies?

- what do we know from animal studies?

- what do we know from human studies?

Where is there evidence that allows us to extrapolate from animals to humans and where should we cautious about extrapolating from animals to humans?

The translational studies from animals to humans are largely disappointing on dietary fibre / SCFA – it would be insightful to try and dissect out where the gaps in our understanding are.

Histone Deacetylases (HDACs).

Much of the detail in this section is quite generic and could be covered in reference to a general review or text on HDAC. Whilst focussed on butyrate mainly (where most evidence of HDAC activity exists) there is no discussion of other SCFA. For example, what role can gut derived acetate play in both supplying acetyl units for histone acetylation and acting as an HDAC inhibitor? Also since butyrate is mainly metabolised in the colonocyte whereas propionate and butyrate are not, what is the significance of HDAC beyond the gut in relation to SCFA?

The sections on SCFAs in Inflammation and SCFAs in Glucose would especially benefit from dissection of animal data and human data. There are methodological (e.g. oral SCFA feeding in animals) and physiological (e.g. fibre loading in diets) that complicate the mixing and extrapolation of animal to human data. This makes for the impression of a much more complete and clinically effective story for SCFA in humans at least than perhaps the evidence allows at present. Therefore this review would be strengthened by explicitly and critically evaluation what the animal data says, what the humans study days and where there are limitation in extrapolating from one to the other.

Author Response

Dear Reviewer:

Thank you very much for taking the time to review and comment the review manuscript titled “A review of short-chain fatty acids and the association with the signal pathways in inflammation and glycolipid metabolism” (Manuscript ID: Ijms-880444). Your comments are important to the revision and quality improvement of the paper. After receiving the email, we carefully read the comments and suggestions, and tried our best to make detailed modifications in the corresponding positions of the paper. Because there are many revisions caused by language errors, we only mark red on major revisions. The above modification has been agreed by all authors of this paper. And we resubmit our manuscript and hope you could consider this paper publish on International Journal of Molecular Sciences. We hope it can meet your suggestions, and consider publishing!

Thank you again for your careful review of this manuscript. If there is something that we are not doing properly, please give us guidance.

All the best with your work!

All authors

2020.8.14

According to your reviewer's suggestions and comments, the following contents are our modifications and replies:

Comment 1:

I am not sure that glycolipid in the title implies the same as those involved in glycolipid research would expect. Perhaps the title “A review of short chain fatty acids and the association with the signal pathways in inflammation, glucose and lipid metabolism” is more apt?

Response: Thank you very much for your suggestion. We revised the title to “Short-chain fatty acids and their association with signalling pathways in inflammation, glucose and lipid metabolism” based on your suggestion.

Comment 2:

Isobutyric and isovaleric acids are generally not include in the definition of SCFA. SCFA refer to straight chain fatty acids whereas isobutyric and isovaleric are usually defined as branched chain fatty acids.

Response: Thank you for your reminding. As you said, isobutyric and isovaleric belong to branched chain fatty acids rather than SCFAs. We have removed isobutyric and isovaleric in the definition of SCFAs. Thanks again for your reminder.

Comment 3:

It is not quite correct that dietary fibre passes through the upper digestive tract unaffected – there are numerous physico-chemical changes that occur – it would be more accurate to say that dietary fibre passes through the upper digestive tract largely undigested.

Response: Really thanks for your careful reading and professional comment. Dietary fiber is still partly digested in the upper digestive tract (For example: saliva, stomach acid, etc.), although very little. This sentence has been revised to “dietary fibre passes through the upper digestive tract largely undigested and is fermented in the caecum and large intestine by anaerobic microorganisms”.

Comment 4:

In addition, besides the liver received some SCFA through the portal circulation, in which other organs and tissues are also affected- this sentence does not make sense.

Response: Thank you for your reminding. We have brought our mother tongue into our writing, and asked professionals to modify the language and grammar of the manuscript. And we have deleted the meaningless sentences. Thanks again for your reminding.

Comment 5:

SCFAs are metabolized in the colonocytes and surplus part are transported into the portal circulation which used as an energy substrate for hepatocytes – not strictly true – most colonic acetate is oxidised in the peripheral muscle and main sequestration in the liver for acetate is into FA synthesis. Propionate is mostly oxidised in the liver.

Response: Different SCFAs are metabolized in different parts. Butyrate is preferentially used as an energy source by intestinal epithelial cells, propionate is metabolized in the liver (to promote liver gluconeogenesis), and acetate reaches the highest systemic concentration in the blood (acetate supplies acetyl-CoA to the cells of the body) (Morrison & Preston, 2016). What we want to express here is that SCFAs are firstly used in colonocytes, and the rest will be transported throughout the body through the blood circulation. This sentence has been modified to “After supplying colonocytes, the remaining SCFAs are transported through the blood to various parts of the body”.

Comment 6:

The contribution of amino acid fermentation (as opposed to metabolism which could also mean uptake by the microbiota) will depend on dietary intake and the introduction should better reflect that the contribution of protein derived amino acids to SCFA production depends on the protein: fibre ratio of the diet.

Response: We fully agree with your statement in this view. Our incorrect spelling caused this problem. The document we referred says that the fermentation of amino acids will also produce SCFAs (less than 1%). This sentence has been modified to “Although anaerobic fermentation of fibre by intestinal microorganisms is the largest source of SCFAs, SCFAs are also formed as products from peptide and amino acid fermentation (less than 1%)”.

Comment 7:

The statement “Although diet and microbiome are the main factors affecting the production of SCFAs, whereas the genetic background of the host and the colonic environment have important effects.” should be supported by references.

Response: We carefully checked these missing references and added them one by one. Here we have listed two documents for reference. In the tenth reference, the pH of the colonic lumen varies with anatomical site and microbial fermentation of dietary residue. The eleventh reference analyzed the intestinal 16s rRNA microbiome data set of different animals (including Mammalia, Aves, Reptilia, Amphibia, and Actinopterygii) and found that the intestinal microbiome was significantly different. This leads to the difference in the production of SCFAs. These show that the host’s genetic background and intestinal environment will affect the production of SCFAs.

Comment 8

SCFAs are absorbed by colonocytes after them were produced which mainly via H+-dependent or sodium-dependent monocarboxylate transporters. – needs references.

Response: We have added the missing references here. In this article, MCT1 was found in the apical membranes of enterocytes, and it transports SCFAs in an H+-dependent electroneutral manner, but it can also transport lactate and pyruvate. Additionally, the electrogenic sodium-dependent monocarboxylate transporter (SMCT)1 is expressed along the entire length of the large intestine and it can also transfer SCFAs.

Comment 9:

Histone Deacetylases (HDACs).

Much of the detail in this section is quite generic and could be covered in reference to a general review or text on HDAC. Whilst focussed on butyrate mainly (where most evidence of HDAC activity exists) there is no discussion of other SCFA. For example, what role can gut derived acetate play in both supplying acetyl units for histone acetylation and acting as an HDAC inhibitor? Also since butyrate is mainly metabolised in the colonocyte whereas propionate and butyrate are not, what is the significance of HDAC beyond the gut in relation to SCFA?

Response: We have modified the details about HDAC in this section, simplifying the background and cutting directly into the topic. In this part, butyrate has been mostly researched, so we ignore the role of acetate and propionate. Thank you for your remanding to us. We joined the discussion of acetate. Acetate, as a donor of histone acetylation, can promote histone acetylation, but this is only a small part (Bulusu, et al. 2017). In addition, acetic acid as a short-chain fatty acid can also inhibit HDAC (Soliman & Rosenberger, 2011). This shows that the regulation of acetate on HDAC is achieved by promoting acetylation and inhibiting deacetylation. As mentioned above, different SCFAs are metabolized in different parts. Butyrate is preferentially used as an energy source by intestinal epithelial cells, propionate is metabolized in the liver (to promote liver gluconeogenesis), and acetate reaches the highest systemic concentration in the blood (acetate supplies acetyl-CoA to the cells of the body) (Morrison & Preston, 2016). SCFAs, not fully utilized by the intestine, are transported through the blood circulation to various organs of the body to inhibit HDAC activity.

Comment 5:

G-protein coupled receptors (GPCRs)

This section lists a long-list of potential signalling effects and mechanisms of SCFA through FFAR2, FFAR3 and GPR109A and interchanges between animal and human evidence quite seemlessly. However, can the animal data simply be extrapolated to humans given that there may be species differences in GPCR signalling? This section would really benefit from a critical analysis of

- what do we know from cell studies?

- what do we know from animal studies?

- what do we know from human studies?

Where is there evidence that allows us to extrapolate from animals to humans and where should we cautious about extrapolating from animals to humans?

The translational studies from animals to humans are largely disappointing on dietary fibre / SCFA – it would be insightful to try and dissect out where the gaps in our understanding are.

The sections on SCFAs in Inflammation and SCFAs in Glucose would especially benefit from dissection of animal data and human data. There are methodological (e.g. oral SCFA feeding in animals) and physiological (e.g. fibre loading in diets) that complicate the mixing and extrapolation of animal to human data. This makes for the impression of a much more complete and clinically effective story for SCFA in humans at least than perhaps the evidence allows at present. Therefore this review would be strengthened by explicitly and critically evaluation what the animal data says, what the humans study days and where there are limitation in extrapolating from one to the other.

Response: Thank you for your question. Please forgive us for putting these questions together to reply. Regarding the GPCRs mentioned here, as well as the following inflammation, glucose metabolism, etc., we will give a unified explanation here. The understanding of the three levels of research on cells, animals, and humans is very helpful to us. We can fully verify the in vivo results at the cellular level, and if these data are also applicable to people, this finding will be very, very useful! We all define that mice and pigs are model animals, and hope that the data based on these animals will have certain benefits for human health. However, it is still a big question whether the inference of animal data to human data is applicable. We cannot simply infer from animals to humans. We agree with you on this point. We didn't put this part in the manuscript because there is really little research on people. Most of the data on humans come from the analysis of mature cell lines and stool residues. And animal research can now be done relatively in-depth. Most of us have got the same results at the non-human cell level and the animal level (for example, SCFAs can reduce inflammation through GPCRs in vivo and cellular level), and human mature cell lines have also obtained consistent results. But whether SCFAs can play such a role in human body is still worth discussing. Different fibre intakes, differences in gut microbiota, host and other reasons may affect the results. Although the current results showed that they have certain relevance, the inability to verify in humans is the biggest problem. This part of the content we have briefly given in Section 7.

Additionally, it is a very good idea to describe in detail the research of GPCRs at the cell, animal, and human levels. But our review mainly focuses on the signaling pathways of inflammation, glucose and lipid metabolism in SCFAs. As for GPCRs and HDAC, we only simply stated the relationship between SCFAs and them. Of course, we can also describe in detail the role of GPCRs at the three levels, but we think this may be another review.

Round 2

Reviewer 2 Report

The response from the authors to the first review are comprehensive and the addition of the final paragraph to contextualise the limitations of the available evidence in human health is welcome and important. From the latest revised manuscript, I have a few comments on the revised text.

Line 67 – “glycolipid metabolism” is misleading. It should refer to “glucose and lipid metabolism”

Lines 47 – 49 – The modified sentence “Although diet and the microbiome are the main factors affecting the production of SCFAs, the genetic background of the host and colonic environment have important effects[10, 11].” Is still not clear what “genetic background” refers to. It would be clearer given the reference cited [11} to rewrite (for example) “Although diet and the microbiome are the main factors affecting the production of SCFAs, species evolution and colonic environment have important effects.” The issue is that the preceding text is focussed on human physiology and the sentence as written can also be mis-interpreted that human genetic background can influence SCFA production to any great extent.

Lines 122-124 – The sentence “Moreover, in the absence of FFAR3, despite the presence of SCFAs in the body, the effects of FFAR3 on inhibiting liver weight and lipid synthesis and preventing obesity in mice induced by high-fat diet were eliminated” is difficult to understand – please rewrite so that is makes sense. Also do you mean liver fat content as opposed to “liver weight”?

Author Response

Dear Reviewer:

Thank you very much for reviewing and commenting the review manuscript titled “A review of short-chain fatty acids and the association with the signal pathways in inflammation and glycolipid metabolism” (Manuscript ID: Ijms-880444) again. Your comments are very helpful to improve the quality of the manuscript. After receiving the email, we carefully read the comments and suggestions, and tried our best to make detailed modifications in the corresponding positions of the paper. The three main changes in the manuscript have been marked in red. The modification has been agreed by all authors of this paper. And we resubmit our manuscript and hope you could consider this paper publish on International Journal of Molecular Sciences. We hope it can meet your suggestions, and consider publishing!

Thank you again for your careful review of this manuscript and best regards!

All authors

2020.8.20

According to your suggestions and comments, the following contents are our modifications and replies:

Comment 1:

Line 67 – “glycolipid metabolism” is misleading. It should refer to “glucose and lipid metabolism”

Response: Thank you very much for your reminding. For “Line 67 – glycolipid metabolism” we have modified “glucose and lipid metabolism”.

Comment 2:

Lines 47 – 49 – The modified sentence “Although diet and the microbiome are the main factors affecting the production of SCFAs, the genetic background of the host and colonic environment have important effects[10, 11].” Is still not clear what “genetic background” refers to. It would be clearer given the reference cited [11} to rewrite (for example) “Although diet and the microbiome are the main factors affecting the production of SCFAs, species evolution and colonic environment have important effects.” The issue is that the preceding text is focussed on human physiology and the sentence as written can also be mis-interpreted that human genetic background can influence SCFA production to any great extent.

Response: Thank you very much for your suggestion. Indeed, our description may be misunderstood. The host genetic background is not completely equivalent to the species evolution in the references. According to your suggestion, this sentence has been revised to “Although diet and the microbiome are the main factors affecting the production of SCFAs, species evolution and colonic environment have important effects”.

Comment 3:

Lines 122-124 – The sentence “Moreover, in the absence of FFAR3, despite the presence of SCFAs in the body, the effects of FFAR3 on inhibiting liver weight and lipid synthesis and preventing obesity in mice induced by high-fat diet were eliminated” is difficult to understand – please rewrite so that is makes sense. Also do you mean liver fat content as opposed to “liver weight”?

Response: Thank you very much for your review comments. What we want to express here is that SCFAs can improve liver metabolism through FFAR3 instead of FFAR2. In addition, in the reference we cited, the liver weight and the liver fat content are reduced. We are very sorry that this sentence is not clear. Now, this sentence has been revised to “Moreover, exogenous supplementation of SCFAs can reduce liver fat content and improve liver metabolism by inhibiting the expression of lipid synthesis genes in FFAR3-deficient mice liver but not FFAR2-deficient.”

Once again, thanks for your careful reading and professional evaluation.
